# Prognostic Significance of Oxidation Pathway Mutations in Recurrent Laryngeal Squamous Cell Carcinoma

**DOI:** 10.3390/cancers12113081

**Published:** 2020-10-22

**Authors:** Molly E. Heft Neal, Apurva D. Bhangale, Andrew C. Birkeland, Jonathan B. McHugh, Andrew G. Shuman, Andrew J. Rosko, Paul L. Swiecicki, Matthew E. Spector, J. Chad Brenner

**Affiliations:** 1Department of Otolaryngology–Head and Neck Surgery, University of Michigan, Ann Arbor, MI 48109, USA; mheftnea@med.umich.edu (M.E.H.N.); apurvadb@med.umich.edu (A.D.B.); shumana@med.umich.edu (A.G.S.); arosko@med.umich.edu (A.J.R.); mspector@med.umich.edu (M.E.S.); 2Department of Otolaryngology–Head and Neck Surgery, University of California Davis, Sacramento, CA 95817, USA; acbirkeland@ucdavis.edu; 3Department of Pathology, University of Michigan, Ann Arbor, MI 48109, USA; jonamch@umich.edu; 4Rogel Cancer Center, University of Michigan Medical School, Ann Arbor, MI 48109, USA; pswiecic@med.umich.edu; 5Department of Internal Medicine, Division of Hematology/Oncology, University of Michigan Medical School, Ann Arbor, MI 48109, USA; 6Department of Pharmacology, University of Michigan Medical School, Ann Arbor, MI 48109, USA

**Keywords:** HNSCC, larynx, Nrf2/Keap1, oxidation

## Abstract

**Simple Summary:**

Organ preservation protocols have become first line therapy for the majority of advanced laryngeal cancers. Unfortunately, up to one third of patients will develop recurrent disease requiring salvage surgery. These tumors tend to display aggressive features when compared to primary disease. The aim of this study is to identify genomic alterations associated with poor prognosis in the recurrent setting to guide precision therapy and identify potential targetable pathways. Here we show that mutations in the oxidation pathway, specifically the KEAP1-NFR2 pathway, predict survival in a cohort of patients undergoing salvage laryngectomy.

**Abstract:**

Organ preservation protocols are commonly used as first line therapy for advanced laryngeal cancer. Recurrence thereafter is associated with poor survival. The aim of this study is to identify genetic alterations associated with survival among patients with recurrent laryngeal cancer undergoing salvage laryngectomy. Sixty-two patients were sequenced using a targeted panel, of which twenty-two also underwent transcriptome sequencing. Alterations were grouped based on biologic pathways and survival outcomes were assessed using Kaplan-Meier analysis and multivariate cox regression. Select pathways were evaluated against The Cancer Genome Atlas (TCGA) data. Patients with mutations in the Oxidation pathway had significantly worse five-year disease specific survival (1% vs. 76%, *p* = 0.02), while mutations in the HN-Immunity pathway were associated with improved five-year disease specific survival (100% vs. 62%, *p* = 0.02). Multivariate analysis showed mutations in the Oxidation pathway remained an independent predictor of disease specific survival (HR 3.2, 95% CI 1.1–9.2, *p* = 0.03). Transcriptome analysis of recurrent tumors demonstrated that alterations in the Oxidation pathway were associated a positive Ragnum hypoxia signature score, consistent with enhanced pathway activity. Further, TCGA analyses demonstrated the prognostic value of oxidation pathway alterations in previously untreated disease. Alterations in the Oxidation pathway are associated with survival among patients with recurrent laryngeal cancer. These prognostic genetic biomarkers may inform precision medicine protocols and identify putatively targetable pathways to improve survival in this cohort.

## 1. Introduction

In the era after the VA larynx and RTOG 91-11 trials, organ preservation protocols with radiation (RT) or chemoradiation (CRT) have become first line therapy for many patients with locoregionally advanced laryngeal squamous cell carcinoma (LSCC) [1,2]. Despite the benefit of laryngeal preservation, disease free survival rates range between 27–70%, and up to one third of patients will require subsequent salvage laryngectomy [1,3,4]. Even when recurrence is operable, survival rates among patients undergoing salvage surgery is approximately 50% and around 30% patients will develop a second recurrence [5,6,7,8].

Currently, the majority of prognostic factors in recurrent LSCC have been clinical (recurrent nodal status, comorbidity scores, perineural and lymphovascular invasion, and positive margins) [5,9,10,11,12]. Work by our group aiming to illuminate the genomic landscape of recurrent LSCC led to targeted sequencing of a small subset of patients with recurrent laryngeal carcinoma after primary surgery or CRT [13]. When compared to primary laryngeal tumors in the Tumor Genome Atlas (TCGA), there was an increased frequency of alterations in multiple genes critical in tumorigenesis including *CDKN2A*, *MTOR*, *PIK3CA*, *TET2*, and *TP53* among others. A study by Lee et al. further identified alterations in p53 expression between previously untreated LSCC and matched recurrent tumors after radiation [14]. While these data offer some insight into the nature of recurrent laryngeal disease, clinically actionable implications remain aspirational [15,16,17].

In the era of precision medicine, an understanding of genomic alterations that underlie recurrent LSCC may shed light on the mechanism of their recalcitrant nature, affording the potential for better patient selection, individualized treatment and targeted therapies. Herein, we investigate the potential prognostic significance of highly recurrent genomic alterations identified in the HNSCC TCGA project in a cohort of patients with recurrent LSCC after primary RT or CRT undergoing salvage laryngectomy and confirm these findings using TCGA.

## 2. Results

### 2.1. Cohort Demographics and Clinical Variables Predictive of Survival

Clinical and oncologic characteristics for this recurrent cancer cohort are presented in Table 1. The majority of patients were Caucasian males with a history of tobacco use and a median age of 59 years (range 39–85 years). Seventy-three percent of patients received RT and 27% received CRT for primary treatment. The most common tumor subsites included the supraglottis and glottis. The median time between completion of primary treatment and salvage surgery was 14 months (range 2mo–13yrs). A larger proportion of patients presented with advanced stage recurrent disease compared to primary disease. Median follow up time was 45 months with a range of 1 month to 18 years.

The five-year overall survival (OS) for the cohort was 45% (95% CI 32–57%) and the five-year disease specific survival (DSS) was 70% (95% CI 55–80%). Stratification by initial overall stage, initial T stage, time to recurrence, recurrent subsite, recurrent T stage, recurrent nodal status, and tobacco use was performed. Patients with positive recurrent nodal disease (cN+) had significantly worse five-year DSS compared to patients without nodal disease (cN0) [39% (95% CI 9–69%) vs. 75% (95% CI 60–86%), *p* = 0.03]. cN+ patients trended to lower five-year OS compared to cN0 patients [20% (95% CI 3–47%) vs. 50% (95% CI 36–64%), *p* = 0.08). None of the remaining clinical variables were significantly associated with either DSS or OS in this cohort.

### 2.2. Characterization of High Frequency Genomic Alterations

To test genetic alterations that may be associated with differential prognoses in this cohort, we focused on a list of candidate genes that were altered at >2% frequency in the original Head and Neck TCGA project [18], which included 226 candidate gene biomarkers, ranging from high frequency alterations such as *TP53* and *PIK3CA*, to low frequency alterations such as *HRAS* and *MDM2*. We also included known actionable genes such as *BRAF* and *PDL1* (*CD274*) in the analysis. Thus, we performed library preparation, targeted capture with our custom library, and next generation sequencing to an average sequencing depth >3000X for our 62 patient cohort. Sequencing and quality control data are shown in Appendix A. Of the 226 genes tested in each patient, our analysis identified an average of 4 somatic alterations per tumor, with a range of 0 to 19 alterations identified across the cohort. We then assigned pathogenicity scores to each of the alterations, and noted that many of the alterations (e.g., *CDKN2A, FAT1*, *PIK3CA* and/or *TP53*) were highly likely to have a pathogenic role in the tumor (Appendix A). *CDKN2A*, *KMT2C* and *KMT2D*, *LRP1B*, *NOTCH1*, *NSD1*, *PDE4DIP*, *PIK3CA*, *TGFBR2*, and *TP53* were altered at high frequencies in our cohort, consistent with findings in laryngeal tumors in TCGA [19,20]. Interestingly, three patients (4.8%) had mutations in *AR,* all of which were comprised of missense mutations. Additionally, *TGFBR2* was mutated at a higher frequency in our cohort (12.9%) compared to primary laryngeal tumors in TCGA (2.8%). Half of these *TGFRB2* alterations are predicted to result in loss of function (LOF) with 3/4 of these representing Lys128Serfs*35 and the other Gln166Ter. Copy number variations (CNV) were also determined and are depicted for each pathway using unsupervised clustering (Figure 1, and shown in Appendix A). Similar to findings in TCGA, multiple focal amplifications in tyrosine kinase were seen in our cohort (*ERBB2*, *FGFR1/3*, and *EGFR*) [18]. In contrast to laryngeal tumors in TCGA, our cohort contained recurrent focal copy loss in genes associated with DNA damage repair (*ATM*, *ATR*, and *BRCA2*). There were no significant associations between copy number variation and survival when assessed by individual gene or grouped by pathway.

### 2.3. Predictive Pathways: Mutational Signatures

To assess the relationship between pathway mutation signatures and survival outcomes, we performed a Kaplan-Meier survival analysis stratifying by the presence or absence of specific pathway mutations. Genes were grouped based on pathways as described in the methods and Table 2, (*RTK/PI3K/RAS, NOTCH, DNA Damage, WNT/Differentiation, Head and Neck Immunity, Cell Cycle, Cell Death, TP53, *Differentiation/Stem/Epigenetic*, Other Kinases,* and *Oxidation*). Oncoplots for overall pathway analysis and for breakdown of gene mutations for select highlighted pathway are shown in Figure 2A–C and Appendix A. Patients with mutations in the *Oxidation* pathway had significantly worse five-year DSS [31% (95% CI 5–64%) vs. 76% (95% CI 61–86%), *p* = 0.02] and five-year OS [13% (95% CI 1–42%) vs. 51% (95% CI 36–63%), *p* = 0.01] compared to those patients without mutations in this pathway (Figure 3).

Similarly, patients with mutations in the *Differentiation/Stem/Epigenetic* pathway had a significantly lower five-year DSS [47% (95% CI 25–67%) vs. 84% (95% CI 66%–93%), *p* = 0.005] and five-year OS [23% (95% CI 8–42%) vs. 59% (95% CI 41–73%), *p* = 0.007] compared to patients without mutations in this pathway. Patients with mutations in both the *Differentiation/Stem/Epigenetic* and the *Oxidation* pathways performed the worst with significantly lower DSS [29% (95% CI 4–61) vs. 76% (95% CI 61–86%), *p* = 0.009] and OS [14% (95% CI 1–46%) vs. 50% (95% CI 35–62%), *p* = 0.03] compared to patients with either no mutations in these pathways or with a mutation in just one of these pathways. The presence of mutations within the *HN*-immunity pathway was associated with improved survival. Patients with mutations in the *HN*-immunity pathways had significantly higher DSS [100% vs. 62% (95% CI 46–75%), *p* =0.02] compared to those patients without a mutation in this pathway. Mutations in the *HN*-immunity pathway were not significantly associated with OS.

Multivariate analysis was then performed to identify independent predictors of DSS and OS. As recurrent nodal status was the only significant pre-operative clinical predictor in this cohort, it was included for multivariate analysis. The rate of *Differentiation/Stem/Epigenetic* pathways mutations was higher in the group of patients without mutations in the *HN*-immunity pathway (41%) compared to the rate in patients with mutations *HN*-immunity pathway (18%), suggesting these mutations may be less likely to occur together. Therefore, each of these pathways were compared in separate models. In a multivariate analysis, the presence of mutations in the *Oxidation* pathway remained a significant predictor of DSS (HR 3.2, 95% CI 1.1–9.2, *p* = 0.03) and OS (HR 2.6, 95% CI 1.2–5.9, *p* = 0.02). The *Differentiation*/Stem/*Epigenetic* pathway also remained a significant predictor of DSS (HR 3.8, 95% CI 1.4–10, *p* = 0.009) and OS (HR 2.4, 95% CI 1.2–4.5, *p* = 0.009). As no disease related deaths occurred in the cohort of patients with mutations in the HN-immunity pathway, no hazard ratio is available. In comparing the multivariate model containing only recurrent nodal status to the model with recurrent nodal status and HN-Immunity mutation status, the inclusion of HN-Immunity status did not significantly improve the accuracy of the model (*p* = 0.3).

### 2.4. Transcriptome Signature Analysis Supports Possible Functional Role of Oxidation Pathway

We found a significantly higher proportion of patients who recurred within one year of initial treatment harbored a mutation in the oxidation pathway (23%), compared to patients who recurred after one year (3%), (Chi squared-*p* = 0.02), further supporting the role of the oxidation pathway in treatment failure. Given these findings and previous research showing that the oxidation response plays a critical role in response to radiation and chemoradiation [21,22], we sought to further investigate the significance of this pathway in our cohort of recurrent larynx tumors. A total of twenty-two patient samples met our quality control standards (DV200 > 31% and > 300 ng RNA) to undergo full transcriptome sequencing. We sequenced the 22 libraries to an average depth of 68,019,597.5 uniquely mapped reads per sample (Appendix A). FPKM expression data for each of the samples is shown in Appendix A. These expression data were used to build the previously described Ragnum hypoxia score [23], which has previously been shown to predict oxidation pathway activity. Results are depicted in the heatmap in Figure 4A. We utilized these data to test for an association between alterations in the oxidation pathway and oxidation pathway activity as represented by Ragnum hypoxia scores (Figure 4B). We found a trend towards higher scores in the patients that harbored alterations in the oxidation pathway (0.3 vs. 0.1, *p* = 0.6). Further, samples containing mutations in *KEAP1*, *CUL3* or *NFE2L2* were all associated with positive scores. We subsequently evaluated whether hypoxia scores were predictive of survival in our cohort and found that there is a trend towards worse five-year disease specific survival in patients with high Ragnum Scores compared to patients with low Ragnum scores, (67% vs. 100%, *p* = 0.2, Figure 4C) and likely limited by the low number of samples in the overall RNAseq cohort.

### 2.5. High NRF2 Protein Expression in Recurrent Larynx Specimens

Evaluation of the *Oxidation* pathway revealed that the majority of mutations occurred in genes involved in the *KEAP1/NRF2* pathway. Subsequent evaluation of *NRF2* protein expression revealed a nuclear rather than cytoplasmic staining pattern (Appendix A). Further, we found that the majority of these recurrent samples were strongly positive (43/52) compared to moderately positive (1/52), weakly positive (6/52), or no staining (2/52), suggesting that the pathway is activated not just by mutations, but potentially by other mechanisms as well in these recurrent tumors. Not surprisingly, given the high level of activity of the pathway in the majority of cases, there was no significant correlation between *NRF2* protein expression with oxidation mutation status (*p* = 0.3), *NRF2* mutation status (*p* = 0.5), or hypoxia score (*p* = 0.9) (data not shown).

### 2.6. Validation of Oxidation Pathway in TCGA

We further leveraged sequencing data from TCGA to evaluate the association between oxidation pathway mutation status and survival (DSS and OS) in a cohort of previously untreated head and neck cancer patients (all head and neck cancer subsites). Using the TCGA Head and Neck Squamous Cell (HNSC) data set (https://www.cancer.gov/tcga), we found that the presence of alterations (mutation or copy number variation) in the oxidation pathway predicted worse five-year DSS (55%, 95% CI 44–66% vs. 63%, 95% CI 55–71%, *p* = 0.02), with a trend towards worse OS (40%, 95% CI 29–51% vs. 49%, 95% CI 42–56, *p* = 0.06), Figure 4D. We hypothesized that oxidation pathway alterations may predict pathway activity in this cohort, similar to the trend observed in our cohort. We therefore utilized three hypoxia scores available in TCGA (Buffa, Ragnum, and Winter Hypoxia scores) to investigate the association between pathway alteration and pathway activity as measured by the hypoxia score (note that only the Ragnum score could be calculated on our cohort). We found that the presence of a pathway alteration was significantly associated with higher Buffa (*p* = 0.01), Ragnum (*p* = 0.009), and Winter (*p* = 0.002) hypoxia scores supporting the hypothesis that alterations predict pathway activity (Figure 4E). To further support the association between oxidation pathway activity and survival, we then tested the association of hypoxia score with DSS and OS. As predicted we found that higher hypoxia scores by all three methods (Buffa, Ragnum, and Winter) were associated with worse five-year DSS and OS (Figure 4F). Controlling for overall stage and HPV status, we found that an increase in hypoxia score by one point was associated with a 3.4% (1.5%–5.3%, *p* < 0.001), 2.7% (−0.003%–5.9%, *p* = 0.08), and 2.2% (0.9%–3.5%, *p* = 0.001) decrease in DSS and a 3.1% (1.7%–4.6%, *p* < 0.0001), 2.3% (0.04%–4.7%, *p* = 0.05), and 2.0% (1.0%–3.0%, *p* < 0.0001) decrease in OS for Buffa, Ragnum, and Winter Hypoxia scores respectively.

## 3. Discussion

Recurrent LSCC in the setting of previous radiation or chemoradiation represents an aggressive disease with significant treatment related morbidity [8,24,25,26,27]. While there has been significant focus on understanding the genomic landscape of primary laryngeal squamous cell carcinoma [28,29,30,31,32,33,34], few studies have explored molecular signatures of recurrent disease [13,15,16,17]. Here we present results from a recurrent laryngeal carcinoma cohort with targeted sequencing and transcriptome data that identified multiple pathways predictive of survival independent of clinical variables.

We found frequent *TP53*, *CDKN2A*, and *PIK3CA* alterations and copy gain/amplification in multiple tyrosine kinase receptors, findings which are consistent with data from laryngeal tumors in TCGA [18]. Notably, our cohort had recurrent loss-of-function (LOF) *TGFBR2* alterations occurring at a higher frequency than previously cited [19,20].

To analyze the correlation between genomic alterations and survival outcomes, mutations were grouped based on biologic effect. Our data demonstrate that alterations in the *Oxidation* and *Differentiation/Stem/Epigenetic* and pathways are associated with significantly worse disease specific and overall survival.

As the Oxidation pathway is thought to play a role in treatment resistance [21,22], this pathway was of particular interest. Evaluation of the *Oxidation* pathway reveals mutations in two predominant genes including Kelch Like ECH Associated Protein 1 (*KEAP1*) and Nuclear Factor, Erythroid 2 Like 2 (*NFE2L2* or *NRF2*). These genes are the key regulators in the *KEAP1-NRF2* pathway responsible for responses to oxidative stress [35]. This pathway has been found to be altered in multiple malignancies [35,36,37,38] *KEAP1* is a negative regulator of *NRF2* and mutations lead to accumulation of *NRF2* in the nucleus resulting in increased transcription of pro-tumorigenic genes. Conversely, mutations in *NRF2* are found in the binding site to *KEAP1* making it resistant to negative regulation by *KEAP1* [35]. Mutations in this pathway are well established in lung cancer occurring in a quarter of squamous cell carcinomas and one third of adenocarcinomas [39,40]. More recently, mutations in this pathway have been identified in breast and head and neck tumors [37,38,41,42]. Kim et al. found a mutation frequency of 13% in previously untreated laryngeal squamous cell carcinoma [41]. Dysregulation of this pathway is also thought to result in reduced sensitivity to chemotherapeutics further suggesting this as a rational biomarker in recurrent populations [35]. Our RNAseq data suggest that mutations in the *Oxidation* pathway may predict increased activity as shown by increased hypoxia scores; however, this analysis was limited by the small number of samples with mutations (n = 3). We also found that the majority of our recurrent laryngeal cancer specimens displayed strong *NRF2* staining suggesting this may serve as a predictive biomarker in the primary setting to identify patients who are at elevated risk of failing CRT. To further investigate the role of Oxidation alterations in Head and Neck cancer, we evaluated sequencing data from TCGA. We found that even in a cohort of previously untreated HNSCC of varying sites (larynx and oral cavity), alterations in the *Oxidation* pathway continued to predict survival. Further we leveraged publicly available hypoxia scores to show an association between *Oxidation* alterations and increased pathway activity. Together these data support a role for altered activation of the *Oxidation* pathway specifically the *KEAP1-NRF2* pathway in not only recurrent laryngeal tumors, but also in Head and Neck cancers more broadly, and these results warrant further investigation in additional cohorts.

We were also interested in evaluating for an association between tobacco use and oxidation pathway mutations given the generation of oxygen radicals with smoking. There were no significant associations between current or former tobacco use and oxidation pathway mutation status. The two never smokers were found to have no mutations within genes in the oxidation pathway. Further evaluation of the two never smoker samples revealed one had no genomic mutations while the other had mutations within the immune and Tp53 pathways. Copy number evaluation of these rare tumors (recurrent larynx in never smokers) revealed copy number gain and amplifications in the RTK/PI3K/RAS and NOTCH pathways.

Additional pathways of interest include *Differentiation/Stem/Epigenetic* pathway. Evaluation of this pathway reveals that the predominant mutations are within genes responsible for chromatin regulation including AT-rich interaction domain 1B (*ARID1B*), lysine methyltransferase 2C (*KMT2C*) and lysine methyltransferase 2D (*KMT2D*). *ARID1B* makes up the largest subunit of the BAF complex (mammalian SWI/SNF) and canonical signaling plays a role in normal cellular development and differentiation [32]. However, alterations within *ARID1B* and other subunits of the BAF complex also play a role in multiple malignancies including adenoid cystic, ovarian clear cell, colorectal cancer and, gastric cancer [43,44,45,46,47]. Alterations in *ARID1B* and other chromatin regulators in hepatocellular carcinoma (HCC) predict degree of liver fibrosis and hepatic vein invasion, suggesting these genes may drive poor prognosis in some tumors [48]. *ARID1B* also serves as a potential therapeutic target as in vitro studies have demonstrated reduced cell growth, increased radiosensitivity and improved DNA damage repair with *ARID1B* inhibition [32,49,50]. Further, these effects are most significant in *ARID1B* mutant cell lines, suggesting *ARID1B* as both a potential biomarker and therapeutic target in our cohort.

Finally, alterations in the *HN*-immunity pathway predict improved disease specific survival in Kaplan-Meier analysis. The most common mutations in this pathway were missense mutations in tumor growth factor beta receptor 2 (*TGFBR2*). *TGFB* is a known driver in intestinal, gastric, prostate, clear cell ovarian, and laryngeal malignancies among others [51,52,53,54]. Further, TGFB signaling has been identified as a key mechanism of tumor immune evasion through inhibition of cytotoxic T cells and creation of an immunosuppressive mileu [55,56]. In head and neck malignancies immune evasion is well documented and recent work has established tumor infiltrating lymphocytes as a prognostic biomarker [57,58,59]. As such LOF mutations in TGFB may indicate increased anti-tumor immunity and may serve as a predictive biomarker in in our recurrent larynx cancer cohort. *TGFB* also serves as an additional potential therapeutic target as multiple studies have demonstrated improved efficacy of anti-PDL1 therapy in combination with TGFB inhibition in vitro [60,61].

## 4. Materials and Methods

### 4.1. Cohort Selection

Inclusion criteria stipulated adult patients treated with RT or CRT for primary laryngeal squamous cell carcinoma with recurrence requiring salvage laryngectomy treated at the University of Michigan. Clinical data including patient sex, age, race, primary overall and TNM stages, primary treatment modality, primary and recurrent subsites, time to recurrence, recurrent overall and TNM stages, tobacco use (defined at the time of salvage surgery), ACE comorbidity scores, and margin status were maintained in a prospectively collected electronic database [62]. Sixty-two patients had available pathologic specimens allowing for DNA extraction and targeted exome sequencing as described below. Access to clinical data and formalin-fixed paraffin embedded (FFPE) specimens was approved by the University of Michigan Institutional Review Board (IRB HUM00080561).

### 4.2. DNA Extraction

DNA extraction from FFPE specimens was performed as previously described [13,63]. Areas of tumor were identified by a board certified head and neck pathologist (J.B.M). Tumor and adjacent normal tissue cores were collected and genomic DNA was obtained using Qiagen Allprep DNA/RNA FFPE kit (Qiagen, Hilden, Germany) and quantified using a Qubit as previously described [63]. DNA samples were then submitted to the University of Michigan Advanced Genomics Core for library preparation and sequencing.

### 4.3. Targeted DNA Sequencing

A minimum of 100 ng of genomic DNA from matching tumor and adjacent normal isolations was used for library preparation with the Rubicon DNA Thruplex (Rubicon Genomics, Ann Arbor, MI, USA) according to manufacturer’s instructions. We then performed targeted capture using our Nextera custom capture library containing high density probes that cover the coding exons of 226 genes, (representing 744kb) found to be altered in >2% of tumors from the first HNSCC TCGA data release, or if noted to be mutated in our internally sequencing data. In total, our custom capture library covers ~0.023% of genome and ~2.47% of the exome. Next generation sequencing was then performed on an Illumina HiSEQ4000 (Illumina, San Diego, CA, USA) using paired end 150 nt reads.

### 4.4. Exome Variant Calling

First, we used FastQC v.0.11.5 (Illumina, San Diego, CA) to access the quality of our sequencing reads. TrimGalore-0.4.5 (Babraham Bioinformatics, Cambridge, UK) was used to trim reads containing sequencing adapters and it was not deemed necessary to perform further quality trimming. Next, we used BWA v0.7.15 to align these processed reads to the hg19 reference genome. The mapped reads were then sorted, de-duplicated and indexed using PicardTools v2.4.1, GATK v3.6 was used to run base quality score recalibration and generate clean aligned reads for variant calling. Variant calling was performed using Varscan v2.4.1. First, pileup files were created for each tumor-normal pair in the set using Samtools v1.9. Then, variants were called from these mpileup files using the somatic mode of the variant caller. Goldex Helix Varseq v2.1.0 (Golden Helix, Bozeman, MT, USA) was used to annotate these variant calls and to filter the variants in the introns and intergenic regions. Variants with a minimum of 5 reads supporting the alternate allele in the tumor samples were considered as potential positives.

### 4.5. Copy Number Analysis

Aberration Detection in Tumor Exome (ADTEx) v.2.0 was used to make copy number estimation calls from the pre-processed tumor-normal BAM files. The software assigns five copy number states from 0 to 4 based on its estimated copy number. State 0 stands for a homozygous deletion, state 1 represents a heterozygous deletion, normal copy number is denoted by state 2, while states 3 and 4 correspond to a copy gain and amplification, respectively. The program was also used to generate representative Manhattan plots for each chromosome of each tumor-normal pair and an R script (R v3.4.0) was used to annotate genes associated with each change.

### 4.6. Transcriptome Sequencing

RNA was isolated from FFPE tissues using the Qiagen All prep kit (Qiagen, Hilden, Germany), and samples with DV200 > 30% were submit to the University of Michigan DNA sequencing core for library preparation and sequencing. Briefly, the Illumina TruSeq Stranded Total RNA library prep kit (Cat#: RS-122-2201/2) was used to prepare libraries according to the manufacturer’s recommendations, with a modification that 14 cycles of PCR was performed to amplify the library prior to the final bead purification. The samples were then loaded on to a total of 6 lanes of an Illumina HiSEQ4000 and paired end sequenced to 75nt length.

### 4.7. Transcriptome Quantification

We first assessed read quality using FastQC (v0.11.5), and then used a two-step STAR workflow to map the reads. In step 1, STAR v2.5.3a was used to generate the genome index database against the reference human genome and annotated transcriptome files. In step 2, read mapping was guided by the genome index database. Only uniquely mapped reads were retained by using samtools (v1.2). To then compute FPKM, we used cufflinks (v2.2.1) with default parameters except for “-max-bundle-frags” which was changed to 100000000 to avoid raising of the HIDATA flag at loci that have more fragments than the pre-set threshold. To calculate the Ragnum hypoxia score, we used the quantification method as described [23], and ComplexHeatmap was used to visualize genes from the hypoxia signature.

### 4.8. TCGA Analysis

Publicly available RNAseq data from the most recent HNSCC TCGA project data release (N = 523) was downloaded and analyzed according to the standard pipelines available through cBioPortal [19,20].

### 4.9. Immunohistochemistry

Immunohistochemical staining was performed on the DAKO Autostainer (DAKO, Carpinteria, CA, USA) using liquid streptavidin biotin (LSAB+) and diaminobenzadine (DAB) as the chromogen as previously described [57,58]. Previously created tumor microarrays (TMA) containing our samples were utilized. Each sample had three representative tumor cores included. The de-paraffinized TMA was labeled with the *NRF2* primary antibody (1:200, Nrf2 Antibody (A-10): sc-365949, Santa Cruz Biotechnology, Dallas, TX, USA) for 60 min at ambient temperature after incubation of the section with background sniper (BioCare Medical, Pacheco, CA, USA) for 30 min at ambient temperature. Subsequently 10 mM Tris HCl/1 mM EDTA pH9 epitope retrieval was performed prior to staining. Appropriate negative (no primary antibody) and positive controls (kidney) were stained in parallel. Using previously published scoring schema [64] each sample was evaluated by a board certified head and neck pathologist (J.B.M) and given an intensity score; 0-no staining, 1-minimal staining, 2-moderate staining, 3-strong staining. Each sample was also given a score for percent of total tumor cells staining positive (1 < 25%, 2 = 26–50%, 3 = 51–75%, and 4 = 76–100%). A final score was calculated by multiplying the two intensity and percentage scores together. Final scores were defined as no staining (0), weakly positive (1–4), moderately positive (5–8) and strongly positive (9–12). For statistical analysis samples were categorized as either no stain/weakly positive (0–4) or moderately to strongly positive (5–12).

### 4.10. Statistics

Genes from our custom capture list were grouped into individual pathways as defined in Hallmark and Go-lists curated by MSigDB [65,66], or added to our HNSCC mutated pathway sets by known biological function. The pathways are outlined in Table 2 and include *RTK/PI3K/RAS (HALLMARK_PI3K_AKT_MTOR_SIGNALING), NOTCH (GO_NOTCH_SIGNALING_PATHWAY), DNA Damage (HALLMARK_DNA_REPAIR), WNT (GO_WNT_SIGNALING), HN-Immunity (GO_INNATE_IMMUNE_RESPONSE), Cell Cycle, Cell Death (HALLMARK_APOPTOSIS), TP53, Differentiation/Stem/Epigenetic (GO_REGULATION_OF_GENE_EXPRESSION_EPIGENETIC), Other Kinases*, and *Oxidation* pathways (GO_OXIDATION_REDUCTION_PROCESS). Somatic mutations included missense, splice region variants, stopgain, 3′ UTR variants, 5′ UTR variants, inframe deletions, frameshift variants, inframe insertions, and initiator codon loss. A pathway was considered to be altered if one or more of the corresponding genes contained a mutation. For CNV survival analysis, a pathway was considered to have a variation is any of the genes contained a copy number alteration. CNV and mutation status and survival were considered separately.

Overall survival (OS) and disease specific survival (DSS) were performed using the Kaplan-Meier method and calculated from the time of salvage surgery to last follow up or death [Prism 8 (GraphPad Software Inc; San Diego, CA, USA)]. Comparison of survival outcomes were calculated with log rank analysis and *p*-values < 0.05 were considered significant. Multivariate analysis was performed using a backward selected binary logistic regression model including any clinical or genomic variables found to have a *p*-value < 0.1 on Kaplan-Meier analysis. Multivariate modeling was performed with SPSS version 26 software (IBM; Armonk, NY, USA). For individual gene analyses, Bonferroni correction was utilized.

## 5. Conclusions

This study provides insight into the mutational signatures of recurrent laryngeal squamous cell carcinoma and suggests biomarkers that may serve to help stratify patients undergoing recurrent cancer surgery. Clinical trials are warranted to explore the implications of specific genomic pathway mutations in patient selection and targeted therapy for patients with recurrent laryngeal cancer.

## Figures and Tables

**Figure 1 cancers-12-03081-f001:**
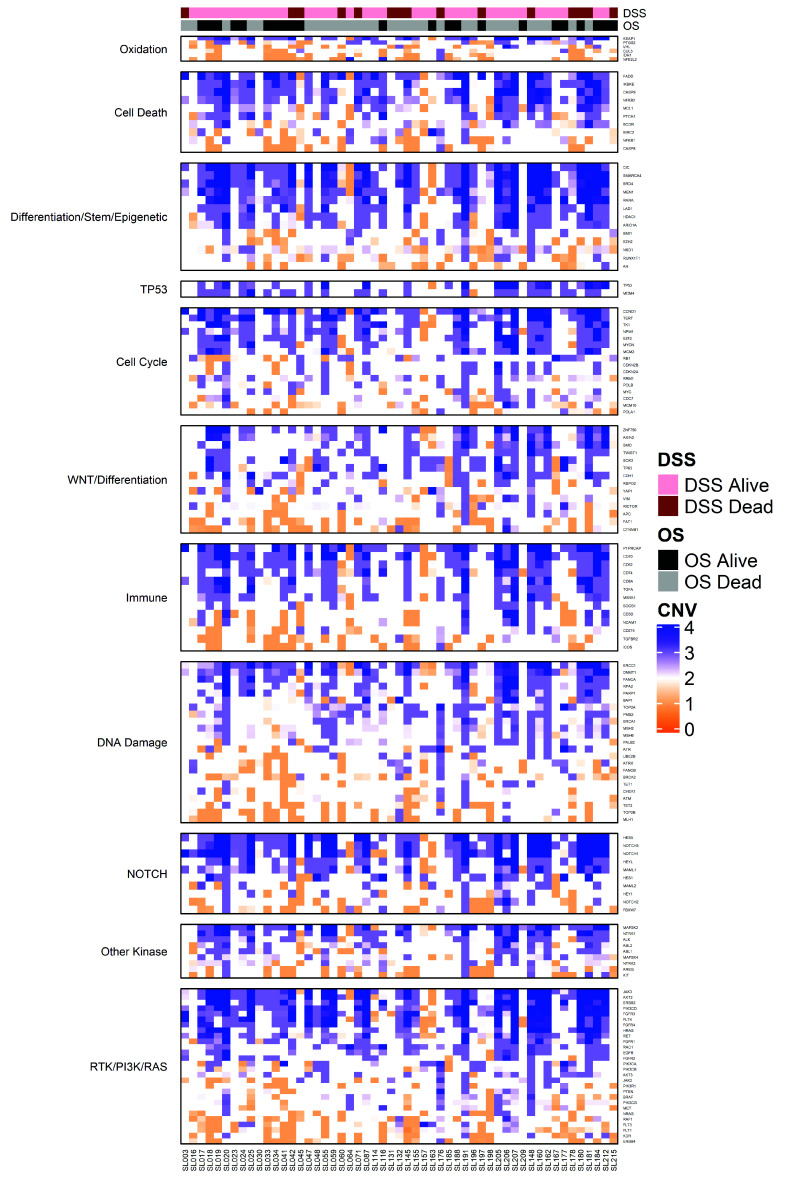
Copy Number Variation (CNV) Analysis. Heat map illustrating unsupervised clustering of CNV analysis grouped by pathway. Disease and Overall Survival status depicted across the top.

**Figure 2 cancers-12-03081-f002:**
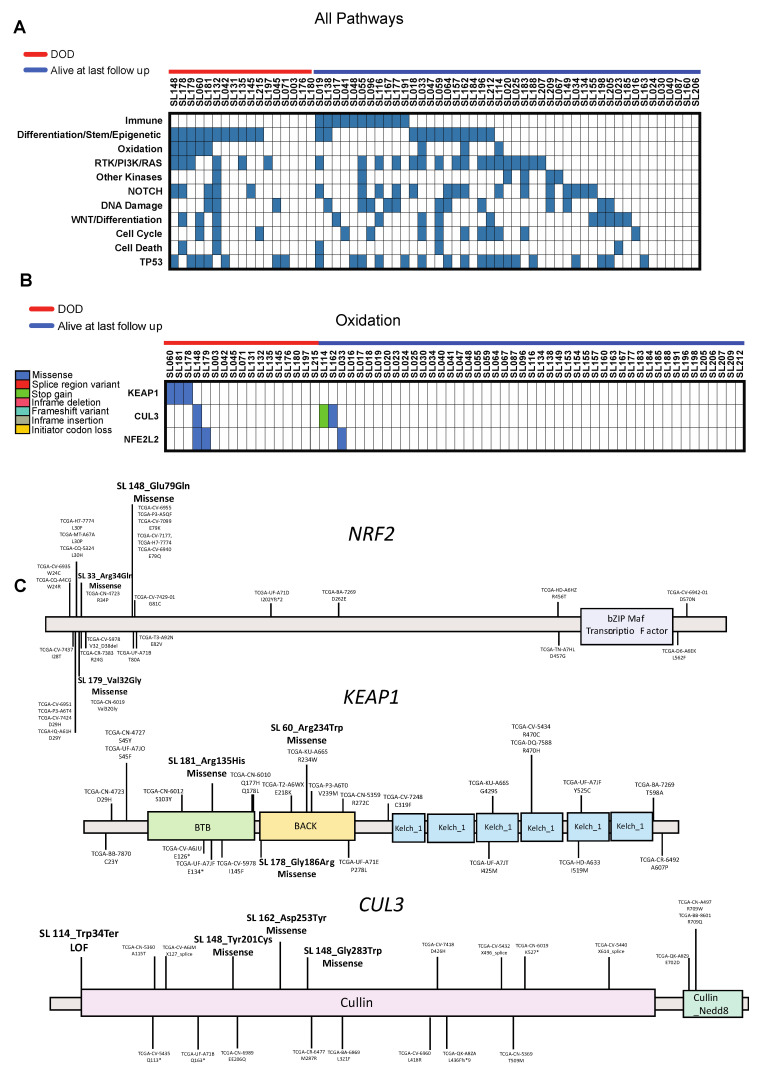
Mutation frequency by pathway and by gene. Oncoplots demonstrating mutations grouped by pathway (**A**) and by gene for Oxidation pathways (**B**). Mutations are clustered by survival (DOD = dead of disease vs. alive) along the top. (**C**) Schematic of mutation location for key genes in Oxidation pathway (*NRF2, KEAP1,* and *CUL3*) in our samples relative to mutations from primary laryngeal tumor specimens from TCGA.

**Figure 3 cancers-12-03081-f003:**
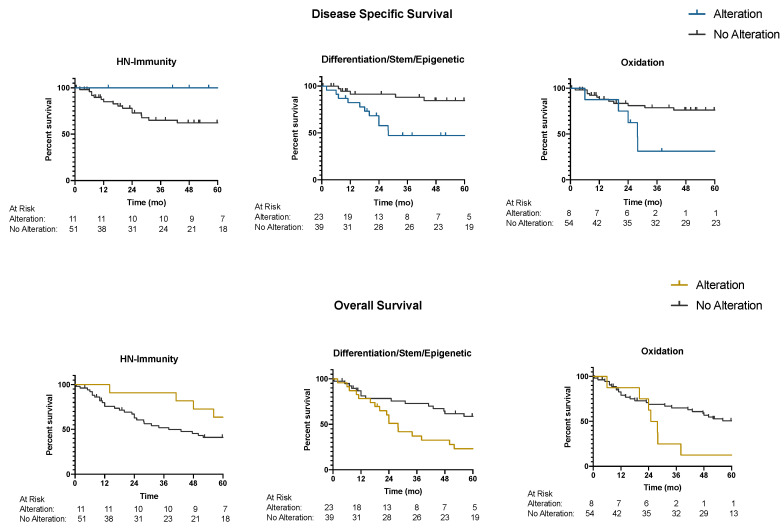
Five-Year disease specific and overall survival by pathway mutation. Kaplan-Meier survival curves for disease specific survival (DSS) and overall survival (OS) are depicted stratifying by mutations in the HN-Immunity pathway (**left**), *Differentiation/Stem/Epigenetic* pathway (**middle**), and Oxidation pathway (**right**). Mutations in the HN-Immunity pathway predict improved DSS and OS, while mutations in the *Differentiation/Stem/Epigenetic* and Oxidation pathways predict worse DSS and OS.

**Figure 4 cancers-12-03081-f004:**
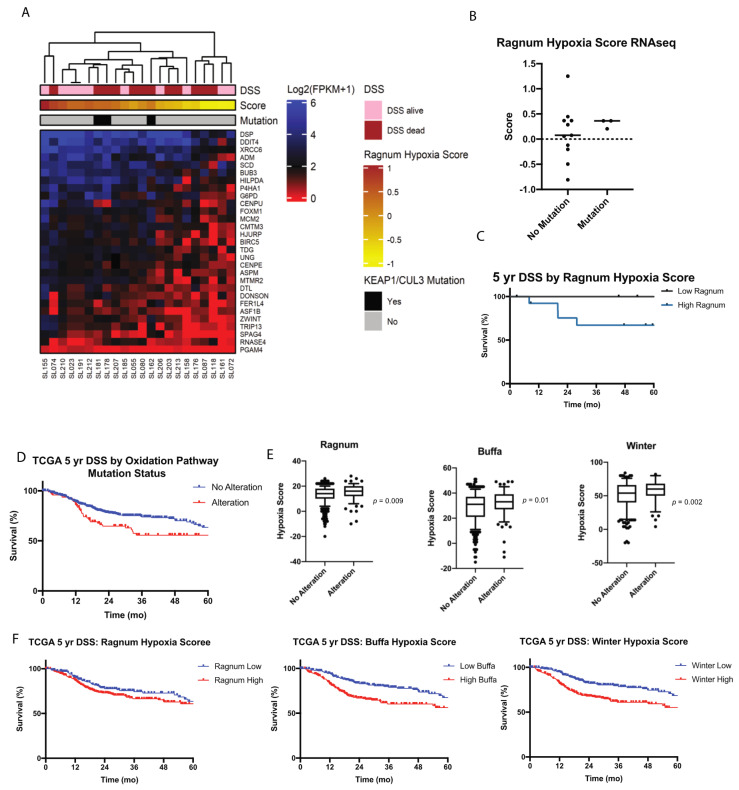
Oxidation pathway analysis in salvage larynx samples and primary TCGA samples. (**A**) Heat map depicting RNAseq data for 22 salvage laryngeal specimens clustered by DSS, Ragnum Hypoxia score, and KEAP1 mutation. (**B**) Ragnum hypoxia score stratified by the presence of absence of mutations in the *Oxidation* pathway. (**C**) Five-year disease specific survival by Ragnum hypoxia score. (**D**) Five-year DSS of TCGA samples stratifying by the presence of alterations in the *Oxidation* pathway. (**E**) Ragnum, Buffa, and Winter hypoxia scores stratifying by the presence of alterations in the *Oxidation* pathways, *p*-value < 0.05 considered significant as tested by Chi-squared. (**F**) Five-year DSS of TCGA samples stratifying by hypoxia scores.

**Table 1 cancers-12-03081-t001:** Demographics.

Variable		Count (%), *n* = 62
Age (yrs)		59 (range 39–85)
Gender	Male	53 (85)
	Female	9 (15)
Race	Caucasian	52 (84)
	Black	3 (5)
	Other	7 (11)
Primary Subsite	Supraglottis	36 (58)
	Glottis	25 (40)
	Subglottis	1 (2)
Primary Overall Stage	I	16 (26)
	II	23 (37)
	III	13 (21)
	IV	7 (11)
Primary T Stage	I	16 (26)
	II	27 (44)
	III	11 (18)
	IV	4 (6)
Primary N Stage	N0	51 (82)
	N+	8 (13)
Treatment	Radiation	45 (73)
	C/RT	17 (27)
Time to recurrence (mo)		14 (2–158)
Recurrent Overall Stage	I	2 (3)
	II	24 (39)
	III	14 (23)
	IV	22 (35)
Recurrent T Stage	I	2 (3)
	II	28 (45)
	III	15 (24)
	IV	17 (27)
Recurrent N Stage	N0	52 (84)
	N+	10 (16)
Recurrent Subsite	Hypopharynx	1 (20)
	Supraglottis	28 (45)
	Glottis	32 (52)
	Subglottis	1 (2)
Tobacco	Never	2 (3)
	Current	35 (56)
	Former	25 (40)

**Table 2 cancers-12-03081-t002:** Pathway definitions.

RTK/PI3K/RAS	Other Kinase	NOTCH	DNA Damage	WNT/Diff.	Immune	Oxidation	Cell Cycle	Cell Death	TP53	Differentiation/Stem/Epigenetic
AKT1	IGF1R	ABL1	DTX1	ATM	MSH2	APC	SMAD2	CUL3	AURKA	MYC	BCL2	HPV16	AR	KMT2C
AKT2	JAK2	ABL2	FBXW7	ATR	MSH6	AXIN2	TGFA	HIF1A	CCND1	MYCL	BCOR	MDM2	ARID1A	KMT2D
AKT3	JAK3	ALK	HES1	ATRX	PALB2	CDH1	TGFBR2	IDH1	CDC7	MYCN	BIRC2	MDM4	ARID1B	LAD1
BRAF	KDR	DDR1	HES5	BAP1	PARP1	CTNNB1	TGIF1	IDH2	CDKN1A	NPM1	CASP8	TP53	ASXL1	MEN1
EGFR	KRAS	KIT	HEY1	BRCA1	PMS2	FAT1	TNF	KEAP1	CDKN1B	POLA1	CASP9		BMI1	NSD1
ERBB2	MET	MAP2K1	HEYL	BRCA2	RPA2	RICTOR	ADA	NRF2	CDKN2A	POLB	FADD		BRD1	RARA
ERBB3	NRAS	MAP2K2	MAML1	CHEK1	TET1	SMO	CD274	PTGS2	CDKN2B	RB1	IKBKE		BRD4	RUNX1
ERBB4	PIK3CA	MAP2K4	MAML2	CHEK2	TET2	SOX2	HLA-A	VHL	E2F1	RRM1	MCL1		CIC	RUNX1T1
FGFR1	PIK3CB	NTRK1	NOTCH1	DNMT1	TOP1	SRC	HLA-B		E2F2	TERT	NFKB1		EZH2	RXRB
FGFR2	PIK3CD	NTRK2	NOTCH2	ERCC1	TOP2A	TP63	HLA-DRA		KNSTRN	TERT_Prom	NFKB2		HDAC1	SMARCA4
FGFR3	PIK3CG	NTRK3	NOTCH3	FANCA	TOP2B	TWIST1	ICOS		MCM10	TK1	PTCH1		KMT2A	
FGFR4	PIK3R1	PTPN11	NOTCH4	FANCB	TRAF3	VIM	LAG3		MCM2	TYMS			KMT2B	
FLT1	PTEN			MLH1	UBE2B	YAP1	MS4A1		MIRLET7C					
FLT3	RAC1					ZNF750	NCAM1							
FLT4	RAF1						PTPRCAP							
HRAS	RET						SOCS1

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
