# Peer review of "Prognostic Significance of Oxidation Pathway Mutations in Recurrent Laryngeal Squamous Cell Carcinoma"

_cancers, 2020, doi:10.3390/cancers12113081_

Round 1
Reviewer 1 Report
Neal et al. have submitted a manuscript whose primary research question is to investigate genomic alterations associated with survival in patients with recurrent laryngeal cancer who were initially treated with nonsurgical treatment. The pt cohort consisted of 62 patients with recurrent laryngeal SCC. In this cohort, pts with mutations in the oxidation pathway (8) independently had worse DSS and OS compared to pts without oxidation pathway mutations. Similarly, pts with mutations in the differentiation/stem/epigenetic pathway (23) also independently had lower DSS and OS, and pts with mutations in both pathways had the worst OS and DSS compared to those with only one or no mutation in these pathways. Mutations in the HN-immunity pathways were associated with better DSS. The authors found a trend toward higher ragnum hypoxia scores in pts with alterations in the oxidation pathway as well as a trend to worse DSS in patients with high ragnum scores. Low numbers did limit the statistical analysis of this study aim. Finally, the authors validated the findings of the oxidation pathway using the TCGA (previously untreated) data, finding that alterations in the oxidation pathway predicted worse DSS and a trend toward worse OS. Recurrent laryngeal cancer is a very important clinical problem and more data are needed to understand how to improve survival in this cohort. The methods and the statistical analyses are well done. Thank you for the opportunity to review, I enjoyed reading this paper, it is very well written.
Below I provide several questions and comments that may help the authors improve the readers’ understanding and interpretation of the findings.
-Looking at clinical variables, can the authors add surgical margin status? It would be interesting to know if any of the genomic alterations were associated with the ability to clear surgical margins.
-Did anyone in the cohort receive re-irradiation after salvage surgery?
-Results, line 76: “There was an even distribution between early and advanced stage tumors for both primary and recurrent tumors.” Please edit this sentence – when looking at primary and recurrent, there seems to be a notable increase in both recurrent T4 tumors and overall stage IV, and fewer T1s and overall stage I.
-Results, line 158: What is the HR for for DSS and OS for mutations in the HN-immunity pathway in the MVA?
-Please define smoking status and the time point at which the data was collected in the methods (i.e. current smokers at the time of salvage treatment?). Smoking during radiation treatment is shown to affect tumor hypoxia. Is smoking during any phase of treatment correlated with alterations in the oxidation pathway?
-Am very interested in know more about the 2 never smokers, who were sequenced. More data are needed in these patients; they are rare. Did you find anything interesting in the sequencing that may be added to the discussion?
-Can the authors better define the cohort from the validation (Results, section 2.6). It appears to consist of previously untreated patients from all subsites and the patients were controlled for HPV status suggesting oropharynx was included. In the discussion, line 261, it states it included larynx and oral cavity. While this section does seem at the periphery of the authors’ main content, due to the major differences between very high risk cohort of recurrent larynx cancer patients that nonsurgical treatment, versus previously untreated HNSCC from all subsites, the authors do a compelling job of explaining the rationale for the inclusion of these analyses in the discussion.
-Table 1 – there is a typo next to hypopharynx
Author Response
-Looking at clinical variables, can the authors add surgical margin status? It would be interesting to know if any of the genomic alterations were associated with the ability to clear surgical margins.
Thank you for this question, we did evaluate the association between pathway mutations and margin status. There were only six patients with positive margins and we found that mutations in the other kinases pathway were associated with having negative margins and thus did not include it. We are happy to include if you feel this strengthens the paper.
-Did anyone in the cohort receive re-irradiation after salvage surgery?
One patient was found to be unresectable at the time of salvage surgery and underwent radiation. No patients received re-irradiation as adjuvant after salvage surgery however there were five patients who received radiation or chemoradiation for a subsequent recurrence, three of which were in the palliative setting.
-Results, line 76: “There was an even distribution between early and advanced stage tumors for both primary and recurrent tumors.” Please edit this sentence – when looking at primary and recurrent, there seems to be a notable increase in both recurrent T4 tumors and overall stage IV, and fewer T1s and overall stage I.
Thank you for this comment, we have edited the sentence to correctly reflect this. “A larger proportion of patients presented with advanced stage recurrent disease compared to primary disease.” (Results – line 76-77)
-Results, line 158: What is the HR for for DSS and OS for mutations in the HN-immunity pathway in the MVA?
This is an excellent question. As there were no events in patients with mutations in the HN-immunity pathway, no HR can to be calculated. An alternative method to evaluate a particular variable effect in multivariable modeling when no events are seen in a group is to evaluate the significance of the model (-2 log likelihood ratio) with and without the variable included. We found no improvement in model significance when including or excluding the HN-immunity pathway, confirming the lack of significance in multivariable modeling (change in model p value p=0.3). We have added the below explanation to the manuscript. Thank you for this edit.
“As no disease related deaths occurred in the cohort of patients with mutations in the HN-immunity pathway, no hazard ratio is available. In comparing the multivariate model containing only recurrent nodal status to the model with recurrent nodal status and HN-Immunity mutation status, the inclusion of HN-Immunity status did not significantly improve the accuracy of the model (p=0.3).” (Results - line 168)
-Please define smoking status and the time point at which the data was collected in the methods (i.e. current smokers at the time of salvage treatment?). Smoking during radiation treatment is shown to affect tumor hypoxia. Is smoking during any phase of treatment correlated with alterations in the oxidation pathway?
This is a critical point, thank you. We have added the definition of smoking status to the methods section (defined at time of salvage surgery). Thirty patients were listed as current smokers at the time of initial radiation therapy. There was no significant association between smoking during radiation and mutations in the oxidation pathway p=0.11. Similarly there was no significant association between current and former/never smokers at the time of surgery with oxidation pathway mutations (p=0.10). We have added a point to the discussion to address this valuable point. (Discussion – line 275-283)
-Am very interested in know more about the 2 never smokers, who were sequenced. More data are needed in these patients; they are rare. Did you find anything interesting in the sequencing that may be added to the discussion?
This is an excellent observation. The two never smokers were examined and one had no mutations identified in the defined pathways while the other had mutations in both the immune and Tp53 pathways. Evaluation of copy number alterations revealed both samples had copy number gain and or amplification of numerous genes in the RTK/PI3K/RAS and NOTCH pathways. Neither had any copy number alterations in genes involved in the oxidation pathway. We have added a sentence to the discussion addressing these rare samples. (Discussion – lines 275-282).
-Can the authors better define the cohort from the validation (Results, section 2.6). It appears to consist of previously untreated patients from all subsites and the patients were controlled for HPV status suggesting oropharynx was included. In the discussion, line 261, it states it included larynx and oral cavity. While this section does seem at the periphery of the authors’ main content, due to the major differences between very high risk cohort of recurrent larynx cancer patients that nonsurgical treatment, versus previously untreated HNSCC from all subsites, the authors do a compelling job of explaining the rationale for the inclusion of these analyses in the discussion.
Thank you for this comment, we agree that this is a more heterogenous population compared to our recurrent laryngeal samples. The TCGA sequencing repository does not separate by subsite (rather it groups as “head and neck”) so this was included as a combined cohort. We have better defined this group in the text (results – line 215) and feel that although it is different from our cohort it still lends support to the overall findings in the paper.
-Table 1 – there is a typo next to hypopharynx
Thank you for this edit, the parenthesis has been added to correct the typo.
Reviewer 2 Report
A very interesting study to identify genomic alterations associated with poor prognosis in recurrent laryngeal cancer. The cohort selection included 4 patients with T4 primary stage (table 1). According to last NCCN guidelines, the main treatment in T4 laryngeal cancer is surgery. It is possible to specify the reasons of a primary conservative treatment in this patients? Typing errors: 1) table 1 -> recurrent subsite -> hypopharynx -> 1 (20 -> lack of round bracket 2) figure 2 -> NFL2 is an error? In table 2 is not in the list of genes and in line 243-244 the predominant genes in oxidation pathway are NFE2L2 OR NRF2
Author Response
The cohort selection included 4 patients with T4 primary stage (table 1). According to last NCCN guidelines, the main treatment in T4 laryngeal cancer is surgery. It is possible to specify the reasons of a primary conservative treatment in this patients?
Thank you for this question. Our institution has previously published a phase II clinical trial on the use of induction paradigms in stage III and IV laryngeal cancer including 32 T4 tumors in which the overall three year survival was 85% (equivalent with historical survival rates) and a 70% laryngeal preservation rate.
Reference: Urba S, Wolf G, Eisbruch A, Worden F, Lee J, Bradford C, Teknos T, Chepeha D, Prince M, Hogikyan N, Taylor J. Single-cycle induction chemotherapy selects patients with advanced laryngeal cancer for combined chemoradiation: a new treatment paradigm. J Clin Oncol. 2006 Feb 1;24(4):593-8. doi: 10.1200/JCO.2005.01.2047. Epub 2005 Dec 27. PMID: 16380415.
Typing errors: 1) table 1 -> recurrent subsite -> hypopharynx -> 1 (20 -> lack of round bracket 2)
Thank you, this has been addresses
figure 2 -> NFL2 is an error? In table 2 is not in the list of genes and in line 243-244 the predominant genes in oxidation pathway are NFE2L2 OR NRF2
Thank you this has been edited to read NRF2